# Local Topography Has Significant Impact on Dendroclimatic Response of *Picea jezoensis* and Determines Variation of Factors Limiting Its Radial Growth in the Southern Sikhote-Alin

Olga Ukhvatkina *, Alexander Omelko and Alexander Zhmerenetsky

Federal Scientific Center of the East Asia Terrestrial Biodiversity, Far Eastern Branch of the Russian Academy of Sciences, 159 100 Let Vladivostoku Avenue, 690022 Vladivostok, Russia; omelko@biosoil.ru (A.O.); zmerenetsky@mail.ru (A.Z.)
* Correspondence: ukhvatkina@biosoil.ru

**Abstract:** Climate change significantly influences forest communities, even leading to their complete transformation. In the case of boreal and temperate forests, it is particularly important to understand how dominant tree species respond to climate changes, as they largely determine the structure of forest communities. In this study, we focus on the Jezo spruce (*Picea jezoensis* (Siebold & Zucc.) Carriere), which is widespread in Northeast Asia. We investigated the climate parameters affecting the radial growth of Jezo spruce and how their influence changes along environmental gradients. For the research, 500 tree cores were collected from 10 sites located at elevations ranging from 460 to 1060 m. We found a negative response of Jezo spruce radial growth to precipitation in July–August and SPEI in July of the current year, maximum temperatures in July–August and November of the previous year. On the other hand, we observed a positive response to the maximum temperatures in January of the current year. Furthermore, we established that the influence of these climatic parameters depends on local topography, with 74.3% of the variance in response values being explained by elevation, slope, and the Topographic Position Index. The results obtained demonstrate that the reaction of Jezo spruce radial growth to climate change will be complex, and the balance between negative and positive effects will depend significantly on local topography.

**Keywords:** dendrochronology; annual ring; tree-ring chronology; Jezo spruce; elevation gradient; environmental gradient



## 1. Introduction

Climate changes influence tree growth and regeneration processes [1–3]. In turn, this can lead to significant shifts in tree species distribution, which is currently being observed and described in numerous studies, e.g., [4–6]. In boreal and temperate zone forests, where the number of dominant tree species is small, the loss of even one species can cause dramatic changes over extensive areas [7]. These changes will be reflected in the forest structure, carbon storage, and potential loss of biodiversity. Therefore, studying the response of tree growth to climate and identifying environmental factors on which this response depends is of critical importance for predicting the functioning of forest ecosystems in the future, as well as for adapting management and sustainable use strategies for forests [8–10].

The Northeast Asia covering an area of approximately 4.5 million square kilometers and hosts some of the best-preserved forest ecosystems in the world [11,12]. It is also one of the most diverse temperate forest areas [11,13]. However, this region is sensitive to climate change [14,15] as it is influenced by the East Asian monsoon and various large-scale oscillations such as El Niño Southern Oscillation, Pacific Decadal Oscillation, and Arctic Oscillation.

The Jezo spruce (*Picea jezoensis* (Siebold & Zucc.) Carriere) is one of the widely distributed tree species in Northeast Asia and an important species for the forest industry [16].

Its range covers northeastern China, Japan, and the Russian Far East, where it is found as far north as 57° N [16–18]. According to Usenko [19], the Jezo spruce occurs in areas with cool and humid air, cannot tolerate the close presence of permafrost, prefers well-drained soils, and avoids stagnant moisture and marshes. Due to its large tree size (height up to 35 m and diameter up to 110 cm), long lifespan (up to 500 years), and the scale of natural disturbances that occur after its death (the area of gaps that appear when tree dies reaches 100 m$^2$), the Jezo spruce significantly shapes the structure of forest communities and exerts a significant impact on dynamic processes [16,19,20].

Approximately half of the current distribution range of the Jezo spruce is located within the Sikhote-Alin Mountains, which are situated along the northeastern coast of the Sea of Japan, and stretches for about 1200 km [16–18]. Due to the large latitudinal span of the mountains, which is about 1200 km, the environmental and climatic conditions vary greatly in its different parts. For this reason, in the Northern Sikhote-Alin boreal forests dominate throughout the elevation range. In the southern part, they are only found at higher elevations, with temperate forests occupying the lower and middle elevations [21]. In the boreal forest zone, the Jezo spruce forms dark coniferous fir-spruce forests, while in the temperate forest zone, particularly in Korean pine-broadleaf forests, it is one of the co-dominant species [22,23]. It should be noted that in the Sikhote-Alin, due to the rugged terrain and temperature inversions, favorable conditions for the Jezo spruce may appear in narrow stream valleys even at relatively low elevations [24].

Global climate change within the Sikhote-Alin Mountains is manifested by rapid increases in temperatures during the second half of the year and changes in precipitation regime due to processes in the Pacific Ocean. Winter precipitation decreases, while summer precipitation, on the other hand, increases, but the annual total remains almost unchanged [15,25,26]. It has also been observed that there are indirect manifestations of climate change, such as an increase in the number and intensity of tropical cyclones (typhoons) and their northward advancement [6,27]. According to forecasts, the Southern Sikhote-Alin experiences particularly significant increases in winter temperatures, and changes in the amount and annual distribution of precipitation [15]. Considering that this territory is the southeastern boundary of the Jezo spruce range (with only small spruce forest patches found on Honshu Island's highlands), it can be assumed that this part of the spruce range will undergo the most significant changes due to the shifting climate in the coming decades.

However, information about the specific climatic conditions crucial for the growth of the Jezo spruce is scarce. The most recent studies on the ecology and physiology of the species were conducted in the mid-20th century [28–30]. Recent research [31] has indicated that at the regional level, the distribution of Jezo spruce is primarily caused by precipitation, both in the cold and warm seasons equally. The next important factor is the sum of positive temperatures, represented by the Kira's warmth index. These results need to be further refined at the local level in different parts of the species' range, as the complex mountainous terrain creates various combinations of habitat conditions.

To study the relationships between climate and tree growth at different scales, from local, e.g., [32] to global [33], dendrochronology methods are commonly used. There are a large number of dendrochronological studies showing that the relationship between climate and radial growth varies along environmental gradients. When considering gradients, many studies focus on elevation gradient, since changes in important climatic parameters, such as surface temperature, are associated with elevation [10,34–38]. However, the importance of the precipitation gradient [39–43], and several other factors, such as soil characteristics, tree age, and size, among others [44–46], is also emphasized. Thus, using dendrochronological methods, it is possible to determine how habitat characteristics change and create spatial variability in factors affecting radial growth and the species' sensitivity to climate variability.

The aim of this study was to (a) identify the climatic parameters influencing the radial growth of the Jezo spruce in the mid-mountain belt of the Southern Sikhote-Alin, (b) find

the main environmental factors along gradients of which the influence of the identified climatic parameters changes (i.e., gradually increases or, conversely, decreases as these climatic parameters become more or less important for tree growth under different site conditions), and (c) determine the dependence radial growth response of Jezo spruce on found environmental factors.

## 2. Materials and Methods

### 2.1. Study Area

The study was conducted in the Southern Sikhote-Alin Mountains, within the territory of the Verkhneussuriisky Research Station of the Federal Scientific Center of the East Asia Terrestrial Biodiversity of the Far Eastern Branch of the Russian Academy of Sciences (Figure 1) (44° 02′ N, 134° 12′ E). The topography of the Station's territory is characterized by low mountains with rounded peaks, and their average slope is 20–25°. The minimum and maximum elevations are 460 and 1060 m, and within this elevation range, approximately 70% of the Jezo spruce range in the southern Sikhote-Alin is found. The climate in this area is monsoonal, with approximately 830 mm of precipitation per year and the majority of the rainfall occurring during the summer period (Figure 2). The mean annual air temperature is 0.9 °C [47]. Forest vegetation covers over 99% of the Station's territory, predominantly consisting of Korean pine–broadleaf forests (55%) and dark conifer fir–spruce forests (30%) in terms of area [48]. Considering the terrain, climate, and vegetation, the territory of the Verkhneussuriisky Research Station is typical for the entire mountainous region of the southern Sikhote-Alin.

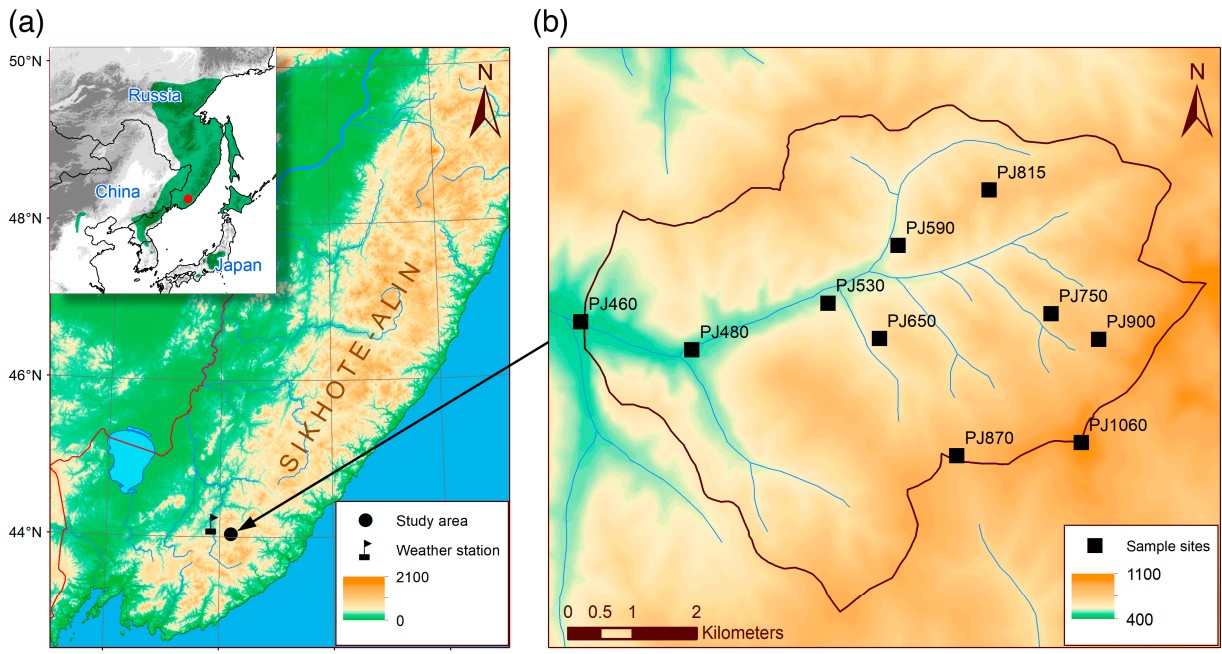

**Figure 1.** Study Area: (**a**) location of the Verkhneussuriisky Research Station and the Chuguevskaya meteorological station in Southern Sikhote-Alin; the green shading in the diagram in the upper left corner shows the distribution of Jezo spruce according to [17]; and (**b**) map of the Verkhneussuriisky Research Station territory showing tree-ring sampling sites.

### 2.2. Tree-Ring Sampling

The data collection was conducted from 2020 to 2022. At the Verkhneussuriisky Research Station, 10 sites of old-growth forests with uniform structure and habitat conditions (area of 0.5–1.5 hectares) were selected, where the Jezo spruce was either the dominant or co-dominant species (Figure 1b). The selection was designed to encompass a maximum elevation range. Forest stands showing signs of anthropogenic disturbances and recent

fires were excluded during the selection process. Within the chosen sites, 1–2 cores (see Table S1 for details) were obtained from each mature Jezo spruce tree (canopy trees with a mean DBH of $34 \pm 5$ cm and a mean height of $25 \pm 2$ m) at a height of 1.3 m. Two cores per tree were initially obtained from the first three sites at the beginning of the study. However, processing these cores revealed that pairs of series of tree ring width measurements with matching periods of sharp growth increase (releases) could negatively affect the quality of chronologies. Therefore, at other sites, we collected only one core per tree. Tree coring was performed perpendicular to the slope and/or tree inclination to avoid sampling compression wood [49]. Between 31 and 68 cores were collected at each site, resulting in a total of 500 cores (Table 1).

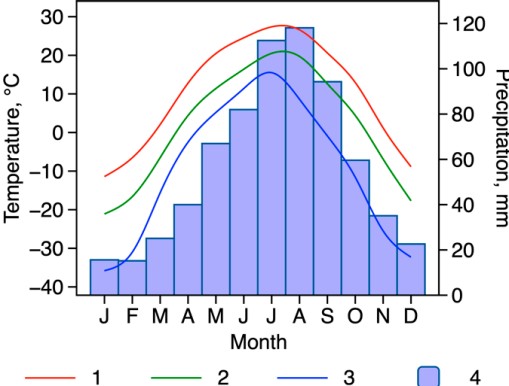

**Figure 2.** Climatogram of the Chuguevskaya meteorological station: 1, 2, 3 denote maximum, average and minimum air temperature, respectively; 4 is monthly precipitation.

### 2.3. Tree-Ring Chronology Development

Preliminary processing of the cores in the laboratory was performed following standard dendrochronological procedures [50]. This process involved mounting, drying, sanding, and increasing the contrast until individual tracheids within annual rings became visible under a binocular microscope. The measurement of annual ring widths was carried out using Velmex semi-automatic measuring system (Velmex INC., Bloomfield, NY, USA), with an accuracy of 0.01 mm.

Subsequently, individual ring width series from each site were crossdated using the TSAP 4.67b software [51]. This involved visually matching individual series with a master series obtained through averaging. The reliability of dating and the detection of missing rings were performed using the COFECHA 6.02 software [52]. Further data processing was conducted using the dplR package [53] in the statistical software R 4.2.1 [54].

To minimize variations in radial growth caused by age-related changes and phytocenotic interactions and to maximize the "climatic" information in the tree-ring chronology, individual series were standardized using a 40-year cubic spline. This standardization method was chosen because Jezo spruce trees growing in the forest experience several periods of sharp growth increase (releases) associated with the formation of canopy gaps [55]. In such cases, standardization using, for example, a negative exponential curve, would fail to remove all such periods. Chronologies were created by averaging the values of individual series using the bi-weighted robust mean [56]. Standard chronologies were used for our investigation.

The following descriptive statistics were calculated to describe the chronologies and assess their quality: the mean sensitivity (MS) [57], RTOT—the mean correlation among ring width series, including correlations between series obtained from the same tree, RWT— the mean correlation between ring width series obtained from the same tree, RBT—the mean correlation among series from different trees, REFF—the weighted mean correlation based on RWT and RBT, EPS—expressed population signal, SNR—signal-to-noise ratio, autocorrelation (AR1), and the standard deviation (SD) [50–59].

**Table 1.** Characteristics of Jezo spruce tree-ring sampling sites. TPI is Topography Position Index [60], TWI is Topographic Wetness Index [61] and ARS is Area Solar Radiation [62,63].

| Site Code | Elevation (m) | Slope (°) | Aspect | Landscape Position | TPI | TWI | ASR (kWh/m²) | Number of Cores (pcs.) |
|---|---|---|---|---|---|---|---|---|
| PJ460 | 460 | 4 | NNE | Foot slope | −0.57 | 8.22 | 1124.21 | 65 |
| PJ480 | 480 | 6 | NNW | Foot slope | −0.29 | 7.89 | 1137.68 | 56 |
| PJ530 | 530 | 6 | NNW | Foot slope | 1.51 | 6.27 | 1028.37 | 68 |
| PJ590 | 590 | 8 | NW | Flood plain | −1.83 | 7.66 | 1110.73 | 33 |
| PJ650 | 650 | 19 | WSW | Narrow ridge | 4.61 | 5.88 | 1170.62 | 31 |
| PJ750 | 750 | 18 | WNW | Side slope | 2.81 | 5.88 | 1139.61 | 63 |
| PJ815 | 815 | 7 | WSW | Narrow ridge | 4.48 | 5.00 | 1030.16 | 57 |
| PJ870 | 870 | 2 | NW | Wide ridge | 1.24 | 6.89 | 1030.16 | 31 |
| PJ900 | 900 | 10 | NW | Wide ridge | 1.63 | 5.59 | 1081.12 | 48 |
| PJ1060 | 1060 | 2 | WSW | Narrow ridge | 4.86 | 5.46 | 1186.16 | 48 |

*2.4. Climate and Topography Data*

The climate data were obtained from the Chuguyevskaya meteorological station (Chuguyevka, 44° 09′ 05″ N, 133° 52′ 10″ E, elevation 260 m), located 30 km west of the study area (Figure 1a). The following climatic variables were used for analysis: monthly precipitation (available data from 1936 to 2019), monthly mean temperature (available data from 1936 to 2019), and maximum and minimum surface air temperatures (available data from 1959 to 2019). It should be noted that meteorological observations were also conducted at the Verkhneussuriisky Research Station starting from 1966 (meteorological station MP7, elevation 800 m, see Supplement, Figure S1). However, these observations were discontinued in 2000 and were continued only in 2017. Therefore, the available continuous observation period covered only 34 years, and meteorological data for the last decades are missing.

Additionally, to analyze the influence of dry and wet years on the radial growth of Jezo spruce, we used the Palmer Drought Severity Index (PDSI) [64] and the Standardized Precipitation Evapotranspiration Index (SPEI) [65]. PDSI data for the period 1901–2021 (0.5° lat-lon resolution) were downloaded from https://crudata.uea.ac.uk/cru/data/drought/ (accessed on 10 March 2023); SPEI data for the period 1901–2020 (0.5° lat-lon resolution) were downloaded from http://sac.csic.es/spei/database.html (accessed on 23 March 2023). In both cases, we used the values of the 0.5° grid cell, the center of which was closest to the study area. The study period was set from 1956 to 2019 because it covers all series of climate data (precipitation, temperatures, SPEI, and PDSI).

Elevation and slope were determined for each site during tree-ring sampling; the landscape position was determined using a topographic map of the area. The TWI and TPI were computed based on the SRTM90 v. 4 DEM [66] using SAGA GIS 8.5 software [67]. Area Solar Radiation was calculated from the same DEM using ESRI ArcGIS 10.2 software and the Spatial Analyst module (Supplement, Figure S2). For TWI, TPI, and Area Solar Radiation, we used the value of the grid cell closest to the center of each site (Table 1).

*2.5. Statistical Analyses*

The influence of climate on the radial growth of Jezo spruce trees was investigated using correlation analysis with the "treeclim" package [68] in R software 4.2.1. The correlation (Pearson correlation coefficient with confidence intervals estimated by bootstrapping) was analyzed between the values of obtained tree-ring chronologies and the climatic variables for each month from June of the year preceding the growth year to September of the current year [57].

To identify the primary environmental factors affecting the dendroclimatic response of Jezo spruce trees depending on local topography, Redundancy Analysis (RDA) was employed. We performed the interactive forward selection procedure, which allows selecting the minimum set of explanatory variables while explaining the maximum possible

variation [69]. In this analysis, the response variables were the correlations between the tree-ring chronologies and the climatic variables for specific months or periods, while the explanatory variables included elevation, slope, Topographic Wetness Index (TWI) [61], Topography Position Index (TPI) [60], and Area Solar Radiation [62,63].

The most significant explanatory variables were selected based on their individual influence until the *p*-value exceeded 0.05, using the Monte Carlo permutation test with 199 random permutations. The results are presented in a biplot ordination diagram. The same analysis was utilized to understand the similarity/difference in the chronologies based on the degree of influence from the environmental factors. Redundancy Analysis was performed using the "vegan" package [70] for R.

For a quantitative statistical assessment of the similarity between the chronologies, a correlation coefficient matrix was computed for the common period from 1956 to 2019. To visualize the results, the "corrplot" package [71] for R was employed.

Additionally, to determine the relationship between the environmental factors and the dendroclimatic response, multiple regression analysis was used. We built regression models from whole set of candidate predictor variables (elevation, slope, TWI, TPI and Area Solar Radiation) by entering and removing predictors based on Akaike Information Criteria [72], in a stepwise manner until there was no variable left to enter or remove any more. For this purpose, the "olsrr" package [73] for R was used.

## 3. Results

### 3.1. Chronology Statistics

Based on crossdating, a total of 433 cores with the best interseries correlation were selected out of 500, and based on these cores, tree-ring chronologies were elaborated (Supplement, Table S1, Figure S3). The length of the chronologies ranged from 171 to 268 years, while the length of the chronologies from the point where the EPS value exceeds 0.85 was approximately one-third shorter, ranging from 105 to 201 years. The average width of the annual rings varied from 0.84 to 1.62 mm. The average EPS values varied from 0.88 to 0.98 for the analyzed period from 1956 to 2019. The REFF values were relatively high for all sets of standardized dendrochronological series, ranging from 0.27 to 0.38. Interseries correlation values ranged from 0.50 to 0.60, and the mean sensitivity ranged from 0.25 to 0.32. It should be noted that most of the statistics, except AR1 and SNR, varied within relatively narrow limits.

### 3.2. Growth–Climate Response

The analysis of the radial growth response of Jezo spruce to climatic variations revealed which climatic parameters and months (or periods) have the greatest influence (Figure 3). Overall, the radial growth of Jezo spruce was significantly influenced by the maximum temperature of the month, precipitation, and SPEI. Significant correlations with the minimum and mean temperatures of the month were sporadic and did not show consistent patterns. Significant correlations with PDSI were only found in two chronologies.

When analyzing individual months, the radial growth is negatively affected by the maximum temperatures of the previous July and August, as well as November. The maximum temperatures of January in the current year have a positive influence (observed in 7 out of 10 chronologies). It should be noted that positive correlations between the tree-ring chronologies and the temperatures of January are also observed for the mean and minimum temperatures of the month, but they are not significant for mean temperatures and are only significant for three chronologies in the case of minimum temperatures. When combining the maximum temperatures of the previous July and August into one period (i.e., analyzing the correlation of chronologies with the average maximum temperatures of July–August), significant correlations were detected in 8 out of 10 chronologies, and the correlation coefficient values decrease regularly with increasing elevation.

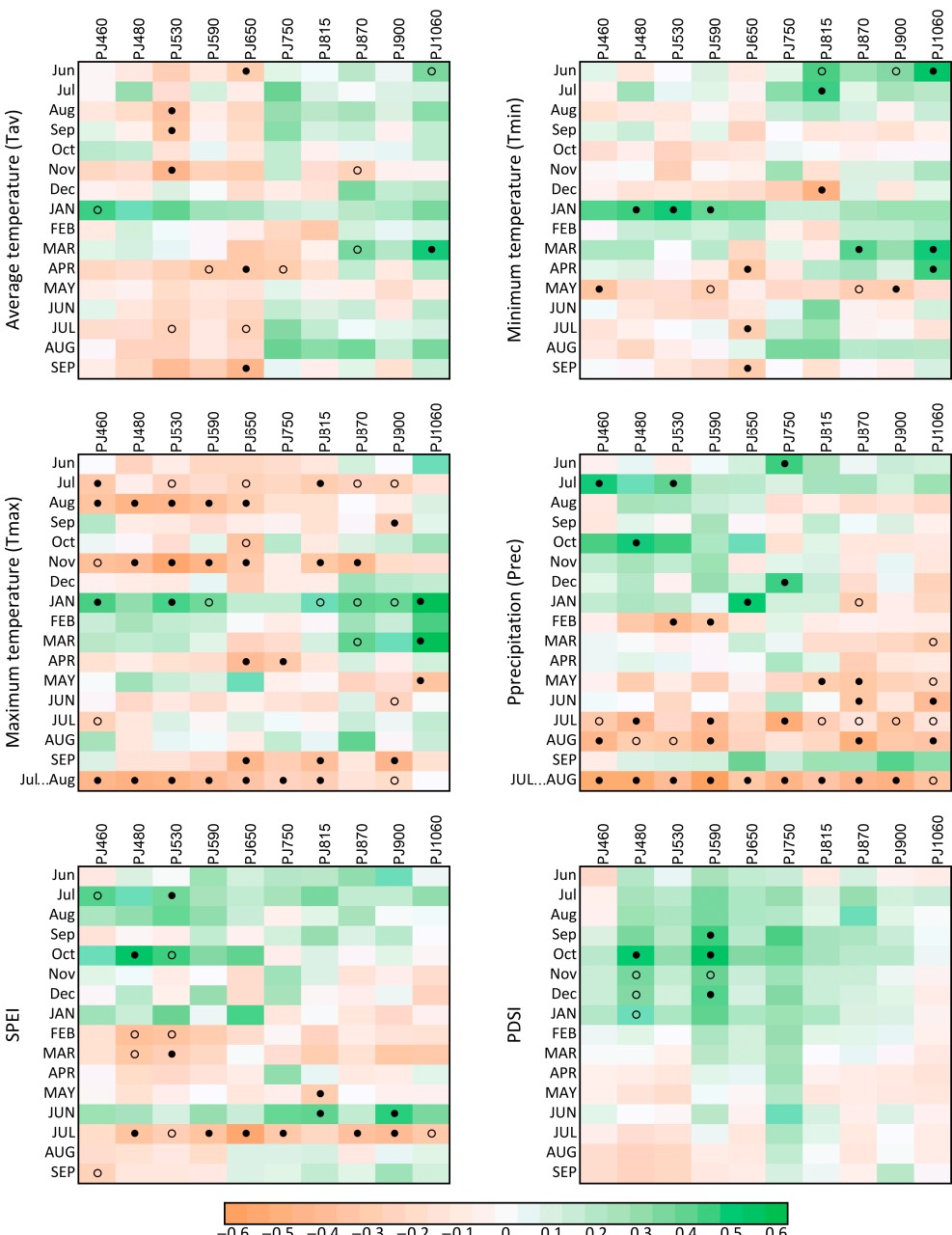

**Figure 3.** Pearson correlation (*r*) between tree-ring chronologies and climate parameters. Capital letters denote the months of the current year; capital and lowercase letters denote the months of the previous year. The values significant at α = 0.5 and α = 0.1 are marked with circles and black dots, respectively.

Additionally, radial growth is negatively affected by the precipitation in July and August of the current year. If we combine them into one period (total precipitation for July–August), then all 10 chronologies significantly correlate with this period. However, unlike the maximum temperatures, we do not observe changes in the correlation values with increasing elevation.

Finally, the 8 out of 10 chronologies negatively correlates with the SPEI in July of the current year, indicating that radial growth is influenced by the ratio between precipitation and evapotranspiration. Interestingly, the influence of SPEI in June is positive, but only two correlation values are significant.

Based on the obtained results, the following climatic parameters, which show the greatest influence on the radial growth of Jezo spruce (significantly correlating with at least

7 out of 10 chronologies) were selected for further analysis: maximum temperatures of previous July–August and November ($Tmax_{Jul...Aug}$ and $Tmax_{Nov}$), maximum temperatures of January of the current year ($Tmax_{JAN}$), total precipitation for July–August of the current year ($Prec_{JUL...AUG}$), and SPEI for July of the current year ($SPEI_{JUL}$).

### 3.3. Redundancy Analysis

The analysis showed that elevation, TPI, and slope were the most significant predictors ($p < 0.05$) of the relationship between climate and the radial growth of Jezo spruce. The included environmental variables accounted for 74.3% ($R^2_{adj} = 61.3\%$) of the variation in climate–growth relationships across sites. The first two RDA axes explained 38.52% and 24.43% of the total variance, respectively (Figure 4).

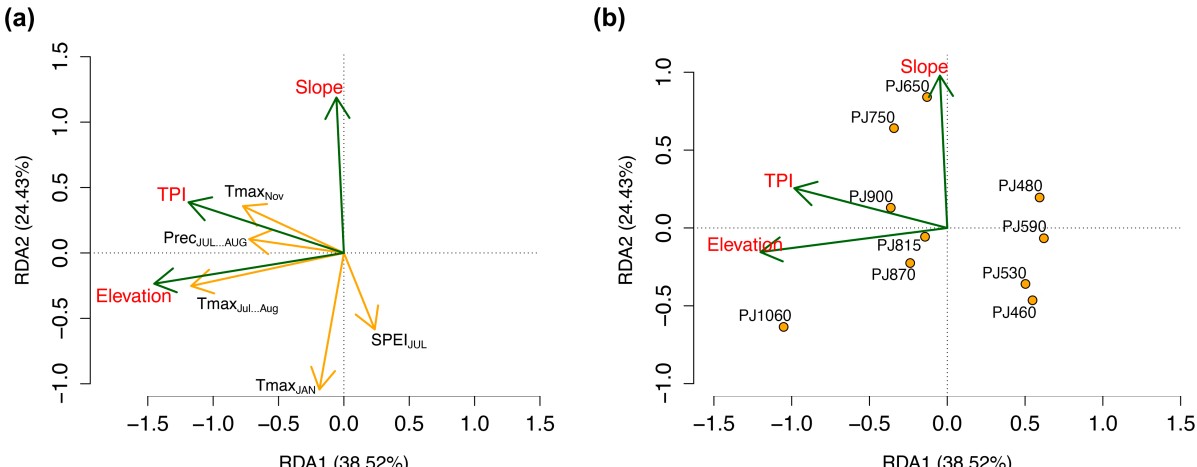

**Figure 4.** (**a**) Ordination biplot of climate–growth correlation coefficients (response variables shown as orange vectors) and environmental variables selected based on the interactive forward selection in RDA ($p < 0.05$). Cosines of angles between vectors approximate the correlation between the variables they represent. Acute, obtuse, and right angles indicate positive, negative, and near-zero correlation, respectively. (**b**) Ordination biplot of sample sites (shown as orange points) and environmental variables (explanatory variables shown as green vectors). The projection of an ordinated point onto a variable vector approximates the variable's value realized for that object.

We found that the negative relationship between radial growth and the mean maximum temperature of the previous July–August was strongly and positively correlated with elevation (Figure 4a). This means that with increasing elevation, the negative influence of July–August temperatures decreased. The negative relationship between growth and precipitation in July–August of the current year positively correlated with TPI. In other words, in valleys (where TPI values are less than 0), the influence of precipitation is greater compared to slopes and ridges (which have high positive TPI values). TPI also positively correlates with the negative relationship between radial growth and the maximum temperature of the previous November.

The positive relationship between growth and the maximum temperature of January of the current year, as well as the negative relationship between growth and SPEI, negatively correlates with slope. This means that on steeper slopes, the positive influence of January's maximum temperatures on tree growth decreases, and at the same time, the negative influence of SPEI increases.

In terms of the included predictors, the sites were divided into four groups (Figure 4b). The first group included four sites with relatively low elevation values (460–590 m) and minimum TPI values. The slope on these sites ranges from 4° to 8°. The second group consists of three sites with average elevation (815–900 m) and TPI values. The slope on these sites ranges from 2° to 10°. The third group comprises two sites with elevation and TPI values similar to the previous group but with significantly steeper slopes (18° and 19°).

Finally, one site stands apart from the others, located at the maximum elevation (1060 m) on a ridge with a slight slope (2°).

### 3.4. Cross-Correlation between Tree-Ring Chronologies

The cross-correlation analysis (Figure 5) revealed that as the difference in elevation between the sites increases, the correlation between tree-ring chronologies systematically decreases. For instance, for the chronologies from sites PJ460 and PJ480 (460 and 480 m above sea level, respectively), the correlation coefficient value was 0.71, while for the chronologies from sites PJ460 and PJ1060 (460 and 1060 m above sea level), it was only 0.37. Ranking the chronologies by TPI values did not reveal any consistent patterns; however, ranking them by slope showed that the two chronologies obtained from sites with a minimum slope of 2° were closely correlated with each other but weakly or insignificantly correlated with other chronologies (see Supplementary Figure S4).

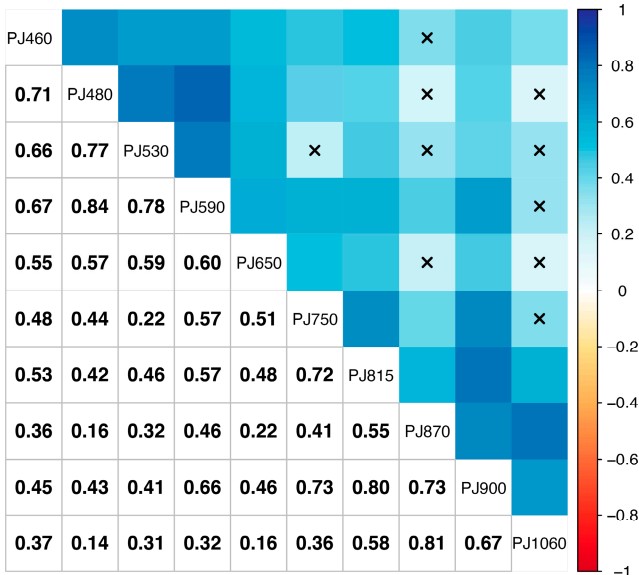

**Figure 5.** Cross-correlation among the tree-ring chronologies. The chronologies are ranked in ascending order based on the elevation of the tree-ring sampling sites. Crosses indicate insignificant correlation coefficients.

The highest significant correlation values were found between chronologies from relatively low (460–590 m above sea level) and high (750–1060 m above sea level) elevations. Interestingly, while RDA showed a distinction of the chronology PJ1060 from the others, this chronology was closely correlated with others obtained from sites with similar elevation values. Additionally, there is a relatively low correlation between the chronologies PJ650 and PJ750, which were distinguished into a separate group by RDA. The chronology PJ750 correlates more strongly with chronologies obtained from higher elevations.

### 3.5. Change in Dendroclimatic Response along Environmental Factor Gradients

The results of the AIC-based model selection using five environmental predictors (factors) are presented in Table 2 and Figure 6. Each equation included only one of the predictors, which, similar to the RDA analysis, were elevation, slope, and TPI. Interestingly, while RDA indicated that $Tmax_{Nov}$ is most strongly correlated with TPI, the multiple regressions analysis revealed a relationship between $Tmax_{Nov}$ and elevation. The proportion of explained variance varied from 0.37 to 0.95, with the highest values observed for the relationship between the dendroclimatic response to the maximum temperatures of the previous July–August and elevation (0.95) and between the response to current July–August precipitation and TPI (0.89). It should be noted that the negative influence of the maximum temperatures of the previous July–August on the Jezo spruce radial growth

increased rapidly with decreasing elevation: the absolute value of the correlation coefficient increased by 0.07 for every 100 m decrease in elevation.

**Table 2.** Regression models showing the relationship between Jezo spruce dendroclimatic response and environmental factors.

| Regression Equation | Parameter | Estimate | SE | p | $R^2$ | $R^2_{adj}$ |
|---|---|---|---|---|---|---|
| $Tmax_{Jul...Aug} = a \times Elevation + b$ | $a$ | 0.00071 | 0.00006 | <0.001 | 0.95 | 0.94 |
| | $b$ | −0.81 | 0.04 | <0.001 | | |
| $Tmax_{Nov} = a \times Elevation + b$ | $a$ | 0.0004 | 0.0001 | 0.002 | 0.39 | 0.31 |
| | $b$ | −0.6 | 0.1 | 0.052 | | |
| $Tmax_{JAN} = a \times Slope + b$ | $a$ | −0.013 | 0.003 | 0.004 | 0.66 | 0.62 |
| | $b$ | 0.39 | 0.03 | <0.001 | | |
| $SPEI_{JUL} = a \times Slope + b$ | $a$ | −0.007 | 0.002 | 0.055 | 0.37 | 0.30 |
| | $b$ | −0.27 | 0.03 | <0.001 | | |
| $Prec_{JUL...AUG} = a \times TPI + b$ | $a$ | 0.035 | 0.004 | <0.001 | 0.89 | 0.87 |
| | $b$ | −0.51 | 0.01 | <0.001 | | |

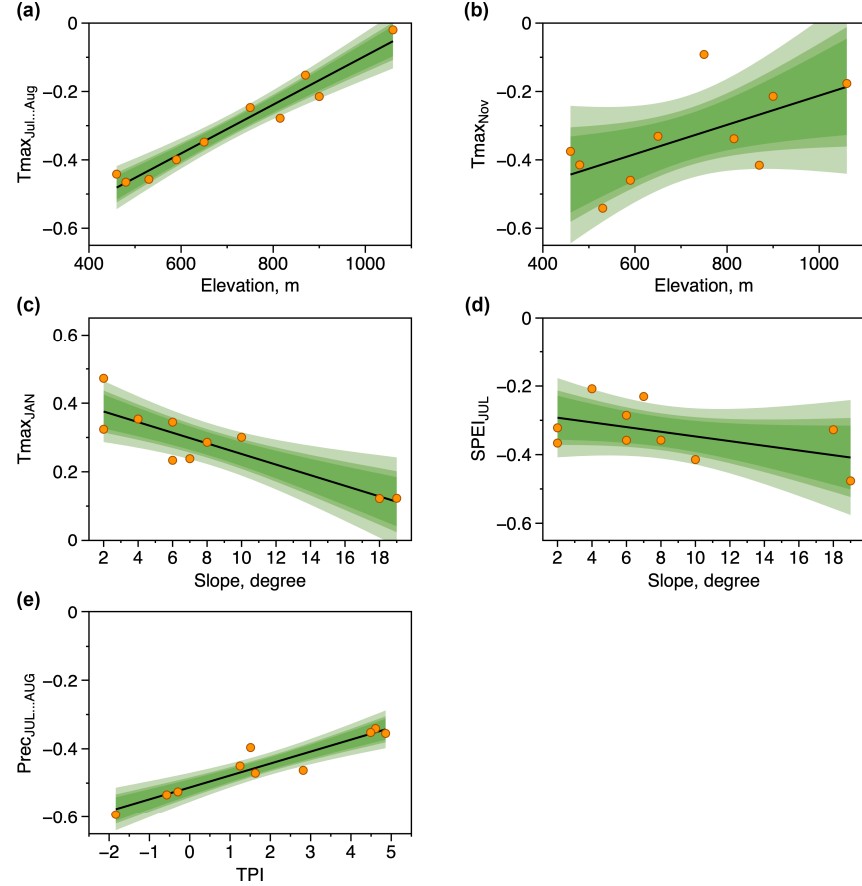

**Figure 6.** Changes in correlation values between tree-ring chronologies and climatic parameters along gradients of the environmental factors. Panels (**a**,**b**) show the relationships between the elevation above sea level and the correlation of the chronologies with maximum temperatures in the previous year's July–August and November, respectively. Panels (**c**,**d**) show the relationships between the slope and the correlation of the chronologies with the current January's maximum temperatures, as well as the SPEI values in the current year's July, respectively. Panel (**e**) shows the relationship between TPI values and the correlation with precipitation in the current year's July–August.

## 4. Discussion

### 4.1. Climatic Parameters Influencing the Radial Growth of Jezo Spruce

The correlation analysis between the tree-ring chronologies and climatic parameters showed that the key climate parameters affecting radial growth are the precipitation during the current July–August and the maximum temperatures (average values of the previous July–August, value of previous November, and value of current January), as well as the ratio between precipitation and evapotranspiration represented by SPEI. For precipitation, maximum temperatures (July–August and November), and SPEI, the correlation with tree-ring chronologies was negative. Positive correlation was found only for the maximum temperatures of the current January.

The relatively weak response of radial growth to the average and minimum monthly temperatures can be explained by the fact that Jezo spruce in the southern Sikhote-Alin region is closer to the southeastern edge of its distribution range, and therefore, its growth is influenced by high temperatures. The insignificant correlation of radial growth with PDSI is related to the fact that Jezo spruce in this region does not experience moisture deficit during the summer period. This is supported by the negative influence of July–August precipitation on radial growth.

Jezo spruce is considered a moisture-loving species [16,19], so the strong negative correlation between the tree-ring chronologies and precipitation was a somewhat unexpected result. The possible reason for this lies in the fact that while Jezo spruce prefers wet and cool habitats, it cannot tolerate waterlogging [16,19]. In the Southern Sikhote-Alin, due to the monsoonal climate with abundant summer rainfall and high humidity (Supplement, Figure S5), Jezo spruce is well supplied with moisture, and an excess of it leads to waterlogging in the soil, negatively affecting the formation of annual rings. This is indirectly supported by the poor growth of the spruce under waterlogged soil conditions [16,19]. During July–August, an average of about 230 mm of precipitation occurs, which accounts for approximately 34% of the annual total. Meteorological observations recorded maximum precipitation sums of up to 474 mm in some years during these months, resulting in severe soil waterlogging. Consequently, abundant summer rainfall influences the mid and late periods of annual ring formation.

In a similar study examining the relationship between the climate and the radial growth of Jezo spruce along an elevational gradient in northeastern China, Changbai Mountains [34], it was shown that the April–May precipitation of the current year negatively affects spruce growth in Korean pine-broadleaf and dark coniferous forests, i.e., the lower and middle mountain zones, whereas precipitation during the entire period of annual ring formation (from October of the preceding year to September of the current year) negatively influences spruce growth in forests with Erman's birch, i.e., the upper mountain zone. The total annual precipitation on Changbai Mountains is 750 mm, which is similar to the amount of rainfall in the southern Sikhote-Alin (800 mm). Moreover, it was shown that even in conditions of relatively low precipitation (200 mm/year) at high elevations (3210–3230 m), winter precipitation can negatively impact the radial growth of *Picea crassifolia* Kom. [35].

The negative response to maximum temperatures in the previous year's summer has been observed in dendrochronological studies of *P. jezoensis*, *P. crassifolia*, and several other species [34,35,74,75]. High temperatures can affect tree growth through various pathways, including increased transpiration accompanied by higher nutrient consumption, the temporary cessation of transpiration (as needle stomata close when temperatures rise to a certain level), and the enhanced evaporation of soil moisture leading to its depletion. In either case, these processes result in a reduction in nutrient reserves, which are built up in the current year and required for the formation of the annual ring in the following year [50,57].

The reasons for the negative response to maximum temperatures in November is apparently more complex. Growth season in the southern Sikhote-Alin region usually concludes in early or mid-September. Before spring, Jezo spruce photosynthesis follows

in such a way that the assimilates produced during this process are not stored [76,77]. Prolonged warm spells, alternating with cold waves during the transition of atmospheric circulation at the boundary between air masses over the continent and the Pacific Ocean in November, lead to the thawing and subsequent freezing of water in the needles, accompanied by the photosynthesis initiation and cessation. This, in turn, leads to the fact that the resources accumulated over the summer are spent on photosynthesis without their replenishment [78]. Consequently, the negative influence of maximum temperatures in July–August and November leads to a reduction in assimilate reserves, which, in turn, results in the formation of a narrower annual ring in the next year [79].

The positive influence of maximum January temperatures in the current year is related to January being the coldest month in the study region. In some years, the average temperature reaches as low as $-45\,^{\circ}\mathrm{C}$, typically averaging around $-18\,^{\circ}\mathrm{C}$ at the Chuguevka weather station and $-21\,^{\circ}\mathrm{C}$ at the MP7 weather station [80]. Overall, the climate in the study area is humid, temperate–cold, and conditions become extremely cold at elevations around 1000 m [81]. Low January temperatures can lead to the compaction of the snow cover, which then melts much slower in spring [82], whereas higher January temperatures result in the softening of the snow cover. If the melting of the compacted snow cover is delayed until the beginning of May, it can lead to a general reduction in the annual growth [82].

The SPEI index takes into account the difference between precipitation and potential evapotranspiration. In July, the average SPEI value in the study area is positive and reaches its highest levels (Supplement, Figure S6), indicating soil moisture accumulation. Therefore, the negative correlation between July SPEI values and tree-ring chronologies, as observed with precipitation, reflects the negative impact of overmoistening on the radial growth of Jezo spruce. This is an interesting result because SPEI is usually used as a drought index, not for overmoistening [83–85]. Despite precipitation being highest in August, the SPEI index reaches its highest positive values in July. In July, the dew point reaches maximum values for the year (Supplement, Figure S5), which means that the air is saturated with moisture and its evaporation is hindered.

In the study area, transpiration activity is significantly influenced not only by precipitation but also by air humidity. During extended rainy days, the transpiration activity of plants is practically zero, whereas on sunny days, a single tree can transpire up to 100 L of water [86]. The absence of transpiration indicates that needle stomata are mostly closed, implying the minimal photosynthetic activity of trees.

*4.2. Local Topography as an Equalizer of the Dendroclimatic Response*

The analysis conducted revealed that the response of Jezo spruce radial growth to maximum temperatures in July–August and November of the preceding year varies with elevation. The response to the total amount of precipitation in July and August of the current year changes with TPI. Finally, the response to maximum temperatures in January of the current year and SPEI values in July of the current year varies depending on the slope. Thus, the variation in the response to different climatic parameters along elevation, TPI, and SPEI gradients affect which climatic parameter limits the growth of Jezo spruce in a particular topographic position.

Before the analysis, we hypothesized that the correlation between chronologies and precipitation in July–August of the current year would be related to the TWI gradient, as this index allows for an assessment of the spatial distribution of soil moisture in the landscape [87–89]. We also assumed that solar radiation could play a significant role in the variation of the dendroclimatic response, as evaporation increases on more insolated slopes [90]. Slope steepness also could be important, as precipitation is retained for a shorter time on steeper slopes. However, the results of the analysis showed that the variation in the correlation coefficient between chronologies and precipitation is better explained by the TPI gradient. The correlation coefficients have the highest absolute values at the minimum TPI values and decrease with increasing TPI values, with a high proportion

of explained variance, reaching 89%. Considering the negative impact of precipitation on the radial growth of Jezo spruce, such a relationship seems logical: the correlation with precipitation has the lowest absolute values for chronologies obtained on narrow ridges, where precipitation is retained for a minimal time, and conversely, the highest values in valleys, where precipitation accumulates. In the study of the growth–climate relationships of *Populus tremuloides* and *Pinus resinosa* along topographic gradients [91], it was also demonstrated that topographic position significantly influences the response of the former species. Moreover, it has been established that topographic position influences tree height growth, particularly of silver fir [92] and Norway spruce [93].

The absolute values of correlation coefficients between chronologies and maximum temperatures (both the mean of previous July–August and previous November) decrease rapidly with increasing elevation. For the tree-ring chronology obtained at an elevation of 1060 m, we did not find significant correlations with the maximum temperatures.

The decrease in correlation values with maximum temperatures at higher elevations is explained by the fact that Jezo spruce is generally adapted to cooler conditions [16,19]. It is also important to note that with increasing elevation, there are changes in seasonal temperature variations. The maximum temperature in August at the Chuguevskaya weather station is 26.7 °C, while at the MP7 weather station, it is 21.3 °C (a temperature difference of 5.4 °C). In November, the maximum temperature at the Chuguevskaya weather station averages 1.3 °C, and at the MP7 weather station, it is 2.54 °C (the temperature difference decreases and is 1.24 °C in absolute value). Moreover, the temperature variation from August to November at the Chuguevskaya weather station is 25.4 °C, while at the MP7 weather station, it is 18.8 °C. Therefore, at higher elevations, not only is it relatively cooler, but there is also a smaller temperature variation during a crucial period for radial growth of Jezo spruce. This is likely one of the reasons why a higher correlation with maximum temperatures is observed at relatively lower elevations.

For the southern distribution boundary of Jezo spruce, a decrease in the negative correlation with maximum temperatures of the previous year's summer was also found with increasing elevation [34], but at higher elevations (from 900 to 1800 m). A decrease in negative correlation with maximum and mean monthly temperatures with elevation has been observed for some other species as well (e.g., [35,38]). However, it should be noted that if the correlation with temperatures in spring or summer is positive, which is characteristic of species at their upper distribution limit, it increases with increasing elevation [10,36].

The positive correlation between chronologies and maximum temperatures of the current year's January decreases with increasing slope. This appears to be logical when considering the slope's aspect: most of the data was collected from sites located on slopes with a northwestern aspect. During the winter period, when the Sun is at its lowest position above the horizon, areas situated on steep northwestern slopes receive relatively little solar radiation. As surface temperature and, consequently, the near-surface air layer temperature are linked to solar radiation [94], the maximum temperatures reached during daytime decrease, thus reducing their positive influence.

The correlation (absolute value) between chronologies and SPEI increases with the slope, meaning that the negative impact of water excess becomes more pronounced as the slope becomes steeper. This may seem counterintuitive, as precipitation tends to accumulate on gentle slopes rather than on steeper ones. However, it is essential again to take into account the aspect of the slopes from which the cores were obtained. Indeed, on one hand, with the slope increasing, the rate of runoff is expected to increase. But, on the other hand, on steeper slopes with a northern and northwestern aspect, solar radiation decreases. Consequently, if the increase in runoff is compensated by a reduction in evaporation, which depends on solar radiation [90], then soil moisture on steeper northern slopes may not decrease compared to gentle slopes but, on the contrary, may increase. In our study, the relationship between the correlation values of chronologies with SPEI and the slope gradient is relatively weak (the absolute value of the correlation increases by 0.035 for

every 5 degrees of slope) and explains only 37% of the variance. Therefore, this conclusion requires further confirmation.

*4.3. Significance of the Obtained Results from the Perspective of Climate Change*

By comparing the correlation values between chronologies and different climatic parameters, we can conclude that in most cases (7 out of 10 sites), precipitation in July and August of the current year is the limiting factor for the radial growth of Jezo spruce trees within the study area. However, for the site located at the highest altitude (PJ1060, 1060 m above sea level), the maximum temperature in current January is the limiting factor. For one of the sites located on the lower part of a slope (PJ530), the limiting factor is the maximum temperature of the previous November. Finally, for the site located on a steep slope (21PJ12), the limiting factor is the ratio between precipitation and evapotranspiration, indicated by the SPEI. The maximum temperatures of the previous July and August are not limiting factor for the spruce growth for any of the sites. Nevertheless, the absolute value of the correlation coefficient between chronologies and these temperatures increases rapidly with decreasing elevation, specifically by 0.07 for every 100 m. This implies that at an elevation of 300 m, the influence of these temperatures will exceed that of other factors, making the temperature of the previous July and August a limiting factor for the radial growth of Jezo spruce trees. Interestingly, at approximately this elevation (200–300 m), the lower boundary of the spruce's distribution in the Southern Sikhote-Alin [16] is found. This can be considered evidence that the lower boundary of its distribution in this region is determined by the maximum summer temperatures.

Currently, the study region is experiencing an increase in temperatures, which is evident both from meteorological observations and dendrochronological studies [25]. According to forecasts, this trend is expected to continue in the near future [15]. Long-term trends in precipitation change based on meteorological observations and dendrochronological studies have not been found in the region [26]. However, there are trends towards aridification with decreasing elevation and, conversely, towards humidification with increasing elevation [95]. It is also essential to consider the increase in typhoon activity, which brings substantial rainfall from July to September [27].

If the existing trends persist, the lower boundary of the Jezo spruce distribution in the Southern Sikhote-Alin will shift upwards due to the impact of maximum summer temperatures. The greatest negative effect of these temperatures in the near future will be observed at elevations between 550 and 650 m, as the correlation between tree-ring chronologies and maximum temperatures in July and August has the highest absolute values for sites at these elevations. Primarily, the participation of the spruce trees in forest stands will decrease on slopes with a southern aspect, where daytime maximum temperatures reach their highest values and where Jezo spruce trees are already relatively rare [48]. At the same time, the humidification of the climate at relatively higher elevations will lead to a decrease in the participation of the spruce in valley forests and on steep northern slopes (where their radial growth is limited by precipitation or the ratio of precipitation and evapotranspiration). As a result, the most favorable conditions for the growth of Jezo spruce will be in the upper parts of mountains and on relatively gentle slopes with a northern or northwestern aspect. In the lower parts of the mountains, the spruce may persist in narrow valleys, where the influence of temperature increases will be less pronounced due to temperature inversions [24]. Thus, local topography not only determines the diversity of climatic parameters that limit the radial growth of Jezo spruce but also influences where future climate changes will have a negative or, conversely, positive impact, resulting in a complex picture of their manifestation.

## 5. Conclusions

Our results complement the findings regarding the influence of habitat conditions, characterized by bioclimatic indices, on the distribution and development of the Jezo spruce [31]. Indeed, we confirmed that, overall, precipitation has a greater impact on the

spruce growth than temperatures. However, we also found that in the Southern Sikhote-Alin, only summer precipitation has a significant influence on growth, and the correlation with precipitation is negative, meaning that excess moisture, rather than deficiency, affects growth. Moreover, the lower boundary of Jezo spruce distribution is determined by maximum summer temperatures (negative influence), while closer to the upper boundary of distribution, the limiting factor for the spruce growth is maximum winter temperatures (positive influence). As a result, the impact of climate change on the radial growth of the Jezo spruce will be complex, and the balance between negative and positive effects, particularly with simultaneous increases in summer and winter temperatures, will largely depend on local topography. Additionally, predicting this impact based only on the current presence or absence of the species on the territory is challenging, as the Jezo spruce occurs throughout the landscape of the study area and surrounding territories.

Furthermore, our results demonstrate that in mountainous areas, where habitat conditions may largely depend on local topography, the same species can be used for reconstructing several climatic parameters even within a relatively small territory. For example, data obtained from site PJ1060 could be used to reconstruct maximum temperatures in January, data from site PJ530 for reconstructing November maximum temperatures, data from site PJ590 for reconstructing precipitation in July and August, and finally, data from point PJ650 for reconstructing the SPEI. This indicates that if a tree growth response to several climatic parameters, it is possible to find local habitats where different climatic parameters limit its growth and to obtain data for a more comprehensive climate reconstruction.

**Supplementary Materials:** The following supporting information can be downloaded at: https://www.mdpi.com/article/10.3390/f14102050/s1, Figure S1: Climatogram of the MP7 meteorological station (Verkhneussuriisky Research Station): 1, 2, 3 are maximum, average and minimum air temperature, respectively, and 4 is monthly precipitation; Figure S2: Topographic variables derived from the digital elevation model (SRTM90): (a) Area Solar radiation (Wh/m$^2$), (b) Topographic Position Index, (c) Topographic Wetness Index; Figure S3: Tree-ring chronologies; Figure S4: Cross-correlation between the tree-ring chronologies: (a) chronologies are ranked by TPI value (from lowest to highest); (b) chronologies are ranked by slope (from lowest to highest). The crosses mark insignificant correlation coefficients; Figure S5: Monthly mean relative humidity and dew point values for the period from 2017 to 2022. Uncertainty bands (blue and orange areas) estimated as twice the standard error of prediction ($\pm 2\sigma$); Figure S6: Monthly mean SPEI values for the study area. Uncertainty bands (blue areas) estimated as twice the standard error of prediction ($\pm 2\sigma$); Table S1: Characteristics of tree-ring chronologies: *MS*—mean sensitivity, *SD*—standard deviation, *AR1*—first-order autocorrelation, *RTOT*—mean correlation between series including correlation between series obtained from the same tree, *RWT*—mean correlation between series obtained from the same tree, *RBT*—mean correlation between series from different trees, *REFF*—weighted mean correlation based on *RWT* and *RBT*, *EPS*—expressed population signal, *SNR*—signal-to-noise ratio.

**Author Contributions:** Conceptualization and methodology, O.U. and A.O.; data collection, O.U., A.Z. and A.O.; data analysis and manuscript writing, O.U., A.Z. and A.O.; review and editing, O.U. and A.O. All authors have read and agreed to the published version of the manuscript.

**Funding:** The study was supported by the Russian Science Foundation, grant No. 22-24-20100, https://rscf.ru/en/project/22-24-20100/ (accessed on 2 October 2023).

**Data Availability Statement:** Data are unavailable due to privacy.

**Acknowledgments:** We thank Ah Reum Han for participation in data collection. We thank the funders for the financial support so that the authors had the opportunity to work on this research. We would like to express our gratitude to the reviewers for their valuable comments and feedback on this manuscript.

**Conflicts of Interest:** The authors declare no conflict of interest.

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
