# Peer review of "Local Topography Has Significant Impact on Dendroclimatic Response of Picea jezoensis and Determines Variation of Factors Limiting Its Radial Growth in the Southern Sikhote-Alin"

_forests, doi:10.3390/f14102050_

Round 1
Reviewer 1 Report
The manuscript is very interesting, the introduction is well founded and the methods to establish the relationship between radial growth and environmental variables and climate are adequate.
The discussion covers the most relevant aspects of the results they obtained.
The comments are minimal, and relate to the following:
- I suggest that objective number two can be a little clearer if explained in more detail.
- In the last paragraph of the discussion: the idea in this paragraph seems to indicate that the local topography is generating climate change. Please modify the wording a little so that it is better understood.
- Some details in the reference list

Reviewer 2 Report
The authors offer a clear and relevant article identifying the links between topography, climate and tree growth. I think that some points of clarification, particularly regarding methodology and results, are necessary before the article can be accepted for publication. However, these corrections do not require any major changes to the manuscript. I will be pleased to read a new version of the manuscript in the coming weeks.
My comments are the following:
# I have several questions regarding sampling: 1) What are the criteria regarding stand age and tree composition in the sampling (lines 128-135)? 2) We also know the number of cores sampled but what is the number of trees sampled, as there can be one or two cores per tree? 3) Also why sometimes only one core was sampled? 4) Apparently, all “mature” trees were sampled within each plot but 31 to 68 cores/trees is not that much for an area of 0.5 ha. What does “mature” represents in terms of DBH, canopy layer or height for example? From the lines 154-161, I imagine that most of these forests were old-growth. Does it means that mostly dominant and codominant trees were sampled?
# Site codes are not very clear to help the readers follow the latitudinal gradient. Could the altitude be used in the site code rather than numbers without any visible order? It will help the reading of the results (e.g., in the section 3.4)
# Using correlations is common in dendrochronological studies, but accumulating correlations often doesn't help identify clear patterns. Talking about significant correlations is also unclear when these correlations can alternate between positive and negative without a clear pattern. It would be interesting if the authors could use a more explicit approach to identify significant results (e.g., more than half the sites with a positive correlation of the same sign). This would make the results much easier to highlight and understand for the reader.
# Part 3.1. It’s unclear why correlations with altitude or other factors are included in this section, which otherwise presents general chronology statistics.
# Lines 49-50 : The sentence “Due to its large tree size, long lifespan, and the scale of natural disturbances that occur after its death” is a bit unclear. Can the authors be more precise about what represent here the words “large”, “long” and “scale of natural disturbance”. These terms may involve different orders of magnitude depending on the ecosystem.
# Lines 58-60: I don’t know the context of the Sikhote-Alin mountains, but here the authors state that Jezo spruce is one of the dominant species of the Korean pine-broadleaf forests, but how can it be dominant if spruce is not cited in this type of forest? Does this mean that Jezo spruce is common, although less common than pines and broadleaf species?
# Line 65: In what direction are going these changes? This word alone is not very informative.
# Lines 121-126: This part is redundant with the lines 58-63 of the introduction
# Table 1: How the landscape position was defined? Is it based on a topographical typology?
# Lines 205-210: It should be explained before the analyses how the topographic variables were extracted.
# Lines 266-267 and 419-420: These phrases are strangely phrased
# Lines 425-427: When does the growing season end in these territories? In November, I imagine the trees have already gone dormant, so how November temperatures can influence tree-ring?
Minor langage editing for some unclear sentences is needed (see my comments)
Reviewer 3 Report
Review report of the manuscript "Local Topography Has Significant Impact on Picea jezoensis Dendroclimatic Response and Determines Variation of Factors Limiting its Radial Growth in the Southern Sikhote-Alin"
The manuscript is presented well but the only concern is about the negative response to the precipitation and SPEI. Though the authors explained the negative response result with other studies, I still feel this is usually inappropriate. The possible reason for this might be related to the chronology development part. The spline method, which is used by the authors, seems not appropriate for the study. Looking at the site description and the raw chronology, I suggest using Friedman's super smoother curve fitting for the development of the chronologies.
Moreover, the existing analysis was carried out using a standard version of the chronology, but the table of the chronology statistics given in supplementary files shows all the chronology has some auto-correlation (AC1). So why authors have not used the residual chronologies for the present analysis?
The overall result is not as promising as the title of the manuscript but can be improved with the suggestion suggested, especially with the development of tree-ring chronologies.
Abstract:
Line 20: that influence/ Make it - that the influence
Line 20: or these/ Make it - of these
Keyword:
Line 25: Remove “Sikhote-Alin” from the keyword as its also mentioned in the title.
Introduction:
Line 43: El Niño / Write full name as El Niño Southern Oscillation
Line 45: add the before forest.
Line 54: Where is Sikhote-Alin Mountain situated? it’s not mentioned anywhere.
Materials and methods
Line 185 & 186: downloaded
Line 188: A short line with an explanation of why the study period was set to 1956–2019 is requested.
Best wishes
Few typos and grammatical error are detected
Round 2
Reviewer 2 Report
I would like to thank the authors for corrections and clarifications to their manuscript. I think that, overall, they address my comments and improve the clarity of the manuscript. I may have two minor comments to add:
- the justification for favoring one core per tree rather than two should be integrated directly into the manuscript so that readers can understand the reasoning and context.
- For Figure 3, wouldn't it be better to reverse the color scale legend from negative to positive?
Other than that, I have no further comments and there will be no need for me to revise a new version of the manuscript. I therefore suggest minor corrections that can be handled directly with the editor. I congratulate the authors on this very interesting manuscript and wish them all the best for their future research.
